# The Impact of Land Transfers on the Adoption of New Varieties: Evidence from Micro-Survey Data in Shaanxi Province, China

**Yi Chen and Zhengbing Wang \***

College of Economics and Management, Northwest A&F University, Yangling, Xianyang 712100, China
\* Correspondence: wzb1964@nwafu.edu.cn

**Abstract:** Land transfers play a vital role in determining the level of farm machinery service and variety selection by scattered land reduction and contiguous land enhancement, which are also conducive to poverty alleviation and welfare utilization. Based on the micro-survey data of 898 kiwifruit growers in Shaanxi Province, this paper analyzed the effect of land transfers on the adoption of new varieties by propensity score matching (PSM) technology. A stepwise regression method was used to test the mediating role of land scale and the moderating role of unmanned aerial vehicles (UAVs). The results suggest the following: (1) Farmers with land transfers had a significant positive effect on the adoption of new varieties, with a 12.5% increase in the likelihood of new variety adoption. (2) The impact of land transfers on the adoption of new varieties was partially mediated through land scale. (3) The positive moderating role of UAV services was empirically emphasized. With the goal of promoting the average income and boosting farmland productivity, the improvement of the land property rights trading market, joint efforts from government and rural cooperatives, and professional and regular training are proposed to optimize land-use structures and reduce machinery service costs, thereby encouraging farmers to adopt new, improved varieties.

**Keywords:** new varieties; land transfers; scale; UAVs

## 1. Introduction

The marginal leveling effect of land transfers is playing an increasingly prominent role in boosting farm productivity and the modernization of farm machinery services, which is considered to be an effective way to pursue the United Nations Sustainable Development Goals (SDGs). The core concern related to land transfers is the application of improved varieties, which makes remarkable contributions to poverty eradication and sustainable production by promoting the fundamental quality and output of agricultural products [1–3]. The adoption of new varieties is also an important part of China's current agricultural supply-side structural reforms, which can alleviate the contradiction between the constantly increasing food consumption demands of residents and the insufficient supply of high-quality, green, and diversified agricultural products in China. With stronger market competitiveness [4–6], the improved new varieties of cash crops can not only adapt to the changing trend of residents' food structures but also promote farmers' income. Many studies in developing countries across the world have provided evidence of the significance of new varieties and the adoption of advanced agricultural technologies in increasing productivity and efficiency [7–11]. However, the actual adoption level of new varieties by farmers is low, which does not match with the renewal rate of new varieties. Meanwhile, as the basis of endowment, which influences the structure of varieties, farmland constrains the labor force input and use of agricultural machinery. With the extensive migration to the city of rural households, land abandonment and fragmentation directly reduce the utility of land, causing the situation to remain grim. Despite the fact that great efforts have been made towards sustainable land use and poverty alleviation, realizing the SDGs is still a long way off.

In resolving these issues, many scholars, such as Le [12], Edilu [13], Bahahudeen [14], and Baiyegunhi [15], have conducted a large number of studies on this area, mainly focusing on the following aspects. First, the factors related to family characteristics and local socioeconomic characteristics could influence the willingness of farmers to adopt similar technologies. The negative impact of age was found in soil-protection technologies [16], whereas the planting scale, educational level, technology extension, and farmers' organization could positively enhance the adoption of irrigation technology to achieve safer vegetable production [17,18]. Second, the adoption intensity, planting structure, and adoption period vary from different households' endowments. Nazli and Smale discovered the factors that encourage changes in variety by applying duration analysis [19]. Farm size was proved to be the vital factor that shortened the time until a farmer replaces one modern variety with another. They tested hypotheses concerning two recurring themes of the Green Revolution: farm size differences and the role of information sources in seed diffusion. They found that time to adoption is only 4 years on average but is shorter on larger farms. Under the influence of different natural conditions, such as climate change, pests, and diseases, varieties differ greatly in terms of yields and stress resistance. Thus, farmers will adopt different varieties and technology combinations to avoid production risks [20]. On one hand, farmers will increase the number of different varieties. On the other hand, they will further avoid risks by allocating different farm sizes to different varieties [21]. Third, the adoption of new varieties can increase food quality, family income, and land productivity [22]. Minten and Barrett advocated that in Madagascar, the communes had higher crop yields, higher real wages for workers, greater food security, and less poverty due to a higher rate of adoption of improved agricultural techniques [23]. Khonje et al. found that the diversification of maize varieties improved farmers' agricultural income and food security by using the micro-data of farmers in eastern Zambia [24]. Data in Tanzania showed that the introduction of new, improved soybean varieties increased household welfare as measured by per capital consumption expenditure [25].

Recently, some studies have focused on the impact of land transfers on the adoption of new technologies and small agricultural machinery services on land cultivation, such as the impact of free farmland transfers on the adoption of conservation tillage technology [26], the adoption of precision agricultural techniques for efficient land administration [27], and the governance assessment of unmanned aerial vehicle implementation in land administration systems [28]. At the moment, land fragmentation and small landholdings per farm household are the major obstacles to farmland operations in China [29]. With the advance of rural market-oriented reforms and the increasing level of the rural–urban migration of farmers, the agricultural land trading market has gradually prospered, and land transfer behaviors have gradually increased. Land transfers refer to the transfer of land-use rights, also known as land contract management rights. There are farmers who transfer small-scale land-use rights to other farmers or economic organizations based on kinship and geography, and there are farmers who transfer land-use rights through land companies. This land reform gives farmers the ability to transfer land to achieve a reasonable match between land, labor, and capital [30,31]. Land in-transfers can directly affect land scale management and improve the efficiency of resource allocation. Di Bu et al. exploited a large-scale land titling reform in China and found that well-defined land property rights increased both land and labor availability and productivity [32]. Farmland transfers have the potential to bring about economies of scale in farmland operations in rural areas [33]. Therefore, appropriately scaled-up agricultural operations are considered to be an important way to promote the adoption of new varieties. Some scholars have investigated the impact of agricultural machinery services on agricultural production from the perspective of the division of the labor force [34,35]. In particular, unmanned aerial vehicles (UAVs) have been widely used in the agricultural field in recent years. For example, UAVs can provide basic flight guidance, flight records, and spray flow control functions, reducing the amount of pesticide drift on non-target crops during the spraying process [36]. Others have out-

lined the prospects and benefits of using unmanned aerial vehicles in different areas of agroforestry, particularly in optimizing pesticide spraying and precise pollination [37].

The preceding research on whether land transfers may assist farmers in the adoption of new varieties has not reached a uniform result. To fill this knowledge gap, this study tests the hypothesis that farmers with land in-transfers would perceive a higher value and have a higher intention of adopting new varieties, which lead to a higher level of adoption behaviors, compared with farmers without land in-transfers. Using field survey data from Shaanxi Province, the main kiwi-producing area in China, PSM technology was adopted to avoid sample self-selection bias in the first stage. Second, intending to explore the mechanisms by which land transfers affect the adoption of new varieties, the intermediate transmission role of land area is empirically described, which emphasizes the importance of scattered land reduction and contiguous land enhancement. Finally, with the advantages of precise and accurate operation, UAV services are introduced to verify the moderating effect, which greatly contributes to green production and sustainable land use.

This research attempts to contribute to this field in terms of the following aspects:

First, the novelty of this study is centered on the integration of land in-transfers-in, the adoption of new varieties, land scale, and UAVs. Although many studies on land transfers [32,33], land scale operations [34,35], the application of UAVs in agriculture [36,37], and the adoption of new technologies have been conducted [22–25], no scholars have combined them into the same conceptual framework. Being different from the current literature, this study generally focuses on the relationship between the land in-transfers and the adoption of new varieties, including the mediating role of land scale and moderating examination of UAVs, which can bridge the research gap and enrich the existing literature to a certain extent.

Second, the findings with regard to kiwifruit of the adoption of new varieties may apply to other cash crops, which is of great practical significance in the implementation of agricultural supply-side reforms. Previous studies mainly focused on food crops, such as wheat [19], maize [24], and rice [10], but little research has been conducted on the adoption and extension of new varieties of cash crops. With excellent performance in the consumer market, the adoption of new varieties of cash crops results in more commercial value-added agricultural products, which not only increases land marginal utility but also narrows the income gap between urban and rural residents by enhancing farmers' earnings.

Third, empirical references and evidence are provided to small households, agricultural enterprise managers, rural cooperatives, governments, and other policymakers by using a representative sample and rational empirical methods. The results highlight the positive impact of land in-transfers on the adoption of new varieties, providing sound guidelines for policy improvements. In resolving the issue of limited farmland resources, the mediating effects of land scale reveal the importance of contiguous land in-transfers. The moderating role of UAVs is confirmed, which prominently contributes to green and sustainable land use. The results also offer an empirical basis for use by some developing countries in Asia, where the major agricultural producers are small-scale households.

## 2. Theoretical Hypothesis

### 2.1. Analysis of the Relationship between Land Transfers and the Adoption of New Varieties

Existing studies, from the perspectives of risk and time preferences [38], scale efficiencies of land use [39], and sustainable land management investments [40], have shown that farmers with larger farms are more inclined to adopt new technologies, which is conducive to the upgrading of scale operations in traditional agricultural production fields. Peng et al. proposed a theoretical model to illustrate the relationship between the scale of land in-transfers and farmers' crop selection behavior. The results show that the size of the land transfer has a U-shaped impact on the market probability of planting food crops. They suggested that the Chinese government should encourage land transfers to guarantee food crop self-sufficiency [41]. Remco H. Oostendorp et al. analyzed the adoption behavior of smallholder farmers by using comparable plot-level duration data for Kenya and the

Philippines. They pointed out that adoption behavior is strongly linked to the process of land ownership transfers, and they argued that policymakers should pay attention to the role of land market dynamics for investment in land [42]. Huaiyu Wang pointed out that the scale of farm production, land ownership, and the market are important factors which determine the frequency and intensity of new variety adoption, and the scale of land has a positive effect on technology adoption behavior [43]. The results also show that factors that encourage changes in variety also differ by farm size, which strongly influences the adoption of precision agriculture technologies [44]. Previous studies, including those on land renting [45], large-scale land acquisitions [46], and the effect of the marketization of urban land transfers on energy efficiency [47], show that land in-transfers can increase farm size, which is positively correlated with the adoption of new agriculture technologies. However, there are few studies on the correlation between land out-transfers and new varieties. The possible reason is that land out-transfers reduce farmers' land resources, whereas the land rent increases part of the farmers' income at the same time. Therefore, there may be both positive and negative impacts on the adoption of new varieties. Based on these findings, this paper generally focuses on the influence of land in-transfers on the adoption of new varieties.

**H1.** *Land in-transfers positively affect the adoption of new varieties by farmers.*

*2.2. Analysis of the Mediating Effect of Land Scale Operations*

Land allocation is one of the basic factors of agricultural production. The direct effect of land transfers is to increase the cultivated land area, and a strong perception of land resource endowments is conducive to farmers carrying out moderate scale management. Meanwhile, due to the increase in production resources, farmers can optimize the planting structure and combination of varieties through rational decision making by adjusting the crop types as well as the planting area of different varieties, so as to avoid risks and obtain greater benefits. Feder et al. showed that the production scale had a positive impact on farmers' agricultural technology adoption behavior [48]. The risk perception may affect farmers' new technology adoption behavior. With the purpose of reducing risks, farmers adjust the planting structure and scale according to the initial endowment [49]. In a study of Sidibe, the relationship between farmland scale and soil and water conservation technology was empirically analyzed. The data showed that farmland size positively affects farmers' technology adoption behavior [50]. In terms of the natural factors of cultivated land, Hu et al. showed that the utilization of machinery services can significantly improve the technical efficiency, with effects varying from province to province [51]. Schuck et al. showed that the larger the land area, the more conducive it is for farmers to adopt more efficient water-saving irrigation technology [52]. In this study, the actual planting scale of kiwifruit was used to represent the land scale of farmers.

**H2.** *Kiwifruit planting area plays a mediating effect on the impact of land transfers on the adoption of new varieties by farmers.*

*2.3. Analysis of the Moderating Effect of UAV Services*

Land transfers reduce the obstacle of cultivated land fragmentation, directly increase the farm size, and make it possible to realize the goal of contiguous land operation, which is conducive to the development of agricultural machinery services. The development and application of agricultural UAV technology is of great significance to accelerate the steps of agricultural modernization in the future, which is also an urgent need for new and efficient agricultural technology in the process of smart agricultural development. In recent years, UAV technology has gradually been used in diversified areas of agriculture, especially in the production process. Many studies, referring to rice production, wheat productivity, corn diseases, and insect damage prevention, have been systematically conducted using UAV spraying technology [53–56]. The selection of high-efficiency spray equipment proved to be

a critical factor in achieving better effects for pesticides and fertilizers [57]. Hu empirically measured the effect of agricultural machinery services on the technical efficiency of wheat production in the framework of the translog stochastic frontier function and Tobit model, and pointed out that the full use of agricultural machinery services could significantly improve the technical efficiency of wheat production. The data also suggest that the degree of improvement varies between different regions [58]. Cai et al. presented an intelligent path-planning method with multiple constraints to solve the problem of the instability of unmanned aerial vehicle intelligent spray-painting systems [59]. The data suggest that pesticide spraying plays a vital role in plant growth regulation and actively participates in inducing various types of stress tolerance. As an effective way to improve wheat yields, an unmanned aerial spraying system can also improve the accuracy of pesticide spraying in the process of pesticide application during the wheat flowering period [60]. Therefore, in order to find the influence of UAV services on land transfers and the adoption of new varieties, the application of UAV pollination and spraying technology in the planting process was used in this paper to represent the service level of small agricultural machines.

**H3.** *UAVs have a moderating effect on land transfers with regard to the adoption of new varieties by farmers.*

Based on the above hypotheses, the conceptual model of land in-transfers on the adoption of new kiwi varieties has been proposed in Figure 1.

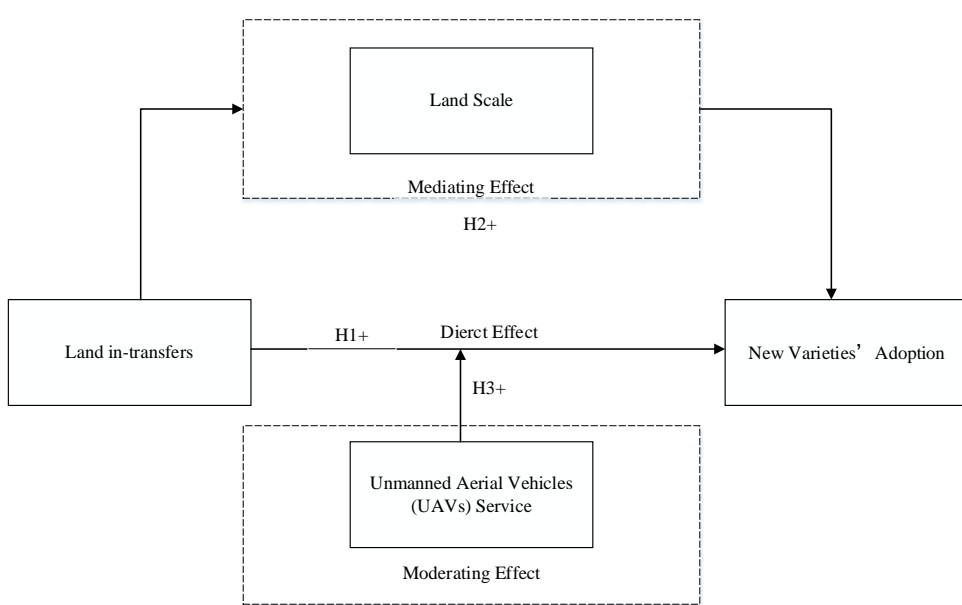

**Figure 1.** The conceptual model of land in-transfers on the adoption of new kiwi varieties.

## 3. Data and Methods

### 3.1. Study Area

As the origin of kiwifruit, China has the largest kiwifruit planting area and highest output of kiwi all over the world, with 2,230,065 tons of output in 2020 [61]. As the main production area of kiwifruit, Shaanxi Province, which is situated in northwest China, had a total planting area of 61,213 hectares in 2020 [62], accounting for approximately 30% of the total planting area in the country. The annual output of kiwifruit in Shaanxi is 1,158,336 tons, accounting for 51.9% of the total, which reflects the status of the kiwifruit industry in China to a large extent [62]. With a width of 50 km from north to south and a length of 300 km from east to west, the main kiwifruit planting zone crosses several regions, namely Mei, Zhou Zhi, and Hu Yi. The location of the study sites is shown in Figure 2, with the kiwi planting area highlighted in blue. Different shades of blue represent different

levels of land transfers in each region, from approximately 13% to 80%, according to the field survey carried out by Northwest A&F University in 2020. The primary occupation of the smallholders is kiwi farming. The adoption of new varieties and the application of new technologies are key parts in the development process of the Shaanxi kiwifruit industry. At present, the varieties of kiwifruit in Shaanxi Province are mainly Xuxiang, Haiwode, and Qinmei. In the kiwi planting belt, land transfers have shown great advantages. Except for agricultural enterprises, the majority of large growers have realized the appropriate scale of operation through land transfers, and most growers' operation areas are between 5 and 15 mu, according to field survey data. At the same time, the flat terrain in the Guanzhong plain and contiguous kiwifruit planting areas provide good basic conditions for UAV operation, which has led this new technology to be rapidly applied in recent years. The cost of UAV operations also shows a trend of gradual decline.

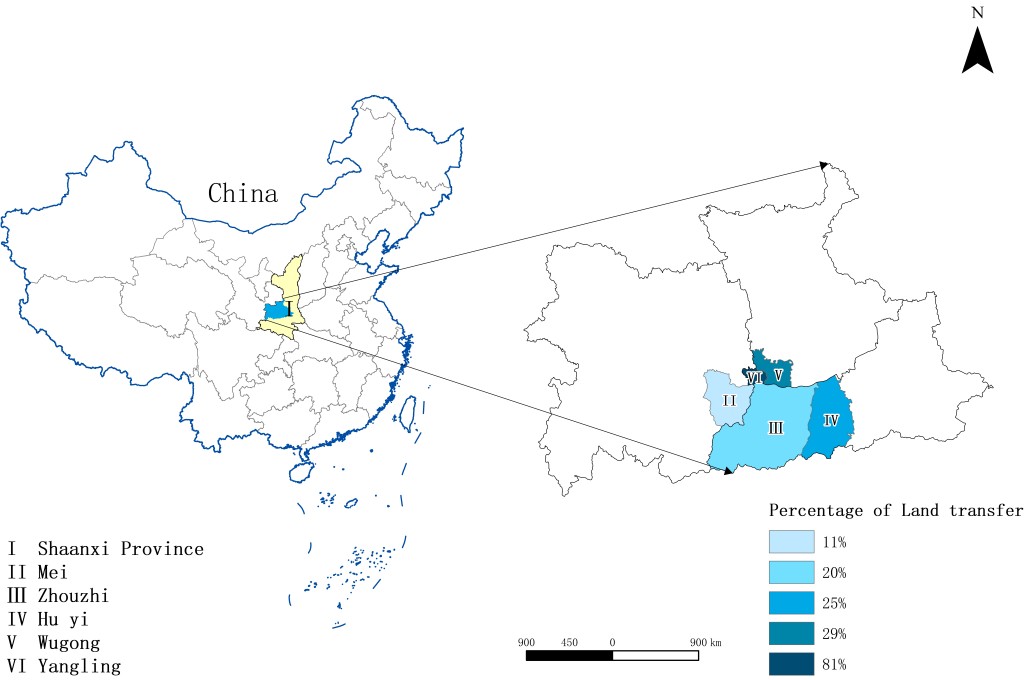

**Figure 2.** Location of the study sites in Shaanxi, China, with the kiwi planting area highlighted in blue.

*3.2. Data Collection*

The field survey of this study was carried out by Northwest A&F University from July to September 2021, and a multistage random sampling technique was employed to select 1000 smallholder kiwi-growing households. In the first stage, five major kiwi-producing districts, namely Mei, Zhou Zhi, Hu Yi, Wugong, and Yang ling, were purposely selected based on the large population of smallholder kiwi farmers in this province. In the second stage, three to five kiwi-growing communities/villages were randomly selected in each of the five districts. Third, smallholder kiwi-farming households were randomly selected from each of the communities/villages, resulting in a total sample of 1000 respondents for the study. These farmers were selected with the help of extension agents who were periodically in touch with them. Primary data were obtained from these sampled respondents through the administration of a well-structured questionnaire consisting of open- and close-ended questions. A total of 1000 questionnaires were distributed, with 950 being collected. Finally, a total of 898 valid samples were obtained after the removal of invalid responses. The field survey found that farmers planted diverse kiwifruit varieties. Other information collected included the personal and family characteristics of respondents, factors of production such as the current situation of variety adoption, land use (land transfers), UAV use, labor, and geographical location.

*3.3. Conceptual and Empirical Framework*

3.3.1. The Benchmark Model

The following benchmark regression (model 1) was constructed, which can analyze the direct impact of land transfers on farmers' variety adoption. Considering that the core explanatory variable of this study, "whether the new varieties have been adopted", is a binary selection variable, it is more appropriate to choose the binary logistic model for analysis.

$$\text{Adoption}_i = \beta_0 + \beta_1 \text{ Transfer}_i + \beta_2 \text{ Control}_i + \varepsilon_{1i} \tag{1}$$

where $\text{Adoption}_i$ represents the status of the adoption of new varieties; if the affirmative answer has been obtained, then $\text{Adoption}_i$ equals 1; otherwise, $\text{Adoption}_i$ equals 0. $\text{Transfer}_i$ refers to the status of land in-transfers; if the affirmative answer has been obtained, then $\text{Transfer}_i$ equals 1, and 0 otherwise. $\text{Control}_i$ represents a set of control variables that may have an impact on the adoption of new varieties, including characteristics of the household head, family, region, and production process. $\beta_0$ is constant, $\beta_1$ is the parameter to be estimated, $\beta_2$ represents the coefficient of the control variable parameter, and $\varepsilon_{1i}$ is the random-error term conforming to a normal distribution.

To avoid sample self-selection bias [63], the PSM technology was applied in this study. The land in-transfer choice may not be random due to factors which may influence farmers' decisions. Therefore, farmers with land-in-transfers may systematically differ from those without land in-transfers because of several factors such as economic and geographical characteristics, which may affect the adoption of new varieties as well. The process of PSM estimation used was as follows: first, a logit model was used to estimate the conditional probability fitting values of samples involved in land in-transfers, which is a set of propensity score (PS) for each variable. Secondly, the appropriate matching method was selected to match the samples of the treatment group (Transfer = 1) and the control group (Transfer = 0) according to the PS value. Finally, the average treatment effect on the treated (ATT) group was obtained.

$$\text{ATT} = E(\text{Adoption}^1 - \text{Adoption}^0 \mid \text{Tansfer} = 1, X = x) = E\,(\text{Adoption}^1 \mid \text{Tansfer} = 1, X = x) - E\,(\text{Adoption}^0 \mid \text{Tansfer} = 1, X = x) \tag{2}$$

In Equation (2), $\text{Adoption}^1$ represents the status of new kiwi varieties' adoption by a farmer who has land in-transfers, and $\text{Adoption}^0$ represents the status of new kiwi varieties' adoption by a farmer who has no land in-transfers.

3.3.2. Mediation Effect Model

In order to further study the impact path of land transfers on the adoption of new varieties, and to test whether there was a mediating effect of land scale operation between land transfers and the adoption of new varieties, the following models (model 2 and model 3) were constructed. Model 2 shows the impact of land transfers on the kiwi planting area. Based on the benchmark, model 3 shows the influence of land transfers on the adoption of new varieties after the introduction of the mediating variable (kiwi planting area).

$$\text{Area}_i = \theta_0 + \theta_1 \text{ Transfer}_i + \theta_2 \text{ Control}_i + \varepsilon_{2i} \tag{3}$$

$$\text{Adoption}_i = \eta_0 + \eta_1 \text{Transfer}_i + \eta_2 \text{Area}_i + \eta_3 \text{ Control}_i + \varepsilon_{3i} \tag{4}$$

In both Equations (3) and (4), the meanings of $\text{Adoption}_i$ and $\text{Transfer}_i$ are consistent with Equation (1). $\text{Area}_i$ is the mediating variable, referring to the actual planting area of kiwifruit. $\theta_0$ and $\eta_0$ are constant terms. $\theta_1$, $\eta_1$, and $\eta_2$ are parameters to be estimated. $\theta_2$ and $\eta_3$ are the parameters of the control variables. $\varepsilon_{2i}$ and $\varepsilon_{3i}$ are the random error terms.

3.3.3. Moderating Effect Model

With the goal of verifying the effect of small farm machinery services on the process of kiwi production, this study introduced UAV Services as the moderating variable to

construct interactive models with land transfers and explored the role of UAV use on the impact of land transfers on the adoption of new varieties of kiwi growers. The process is divided into two parts, including model 4 and model 5. Model 4 shows the influence of the introduction of the moderating variable (UAV Services) on the adoption of new varieties based on the benchmark model. Model 5 shows the impact on the adoption of new varieties when the interaction term (Transfer$_i$×UAV Services) is introduced on the basis of Model 4.

$$\text{Adoption}_i = \alpha_0 + \alpha_1 \text{ Transfer}_i + \alpha_2 \text{UAV Services} + \alpha_3 \text{ Control}_i + \varepsilon_{4i} \qquad (5)$$

$$\text{Adoption}_i = \gamma_0 + \gamma_1 \text{ Transfer}_i + \gamma_2 \text{UAVs Service} + \gamma_3 \text{ Control}_i + \gamma_4 \text{ Transfer}_i \times \text{UAVs Service} + \varepsilon_{5i} \qquad (6)$$

In Equations (5) and (6), where the meanings of Adoption$_i$ and Transfer$_i$ are the same as Equation (1), the moderating variable (UAVs Service) equals 1, referring to pollination or the pesticide/fertilizer spraying of UAVs in the production process, and 0 otherwise. $\alpha_0$ and $\gamma_0$ are constant terms. $\alpha_1$, $\alpha_2$, $\gamma_1$, $\gamma_2$, $\gamma_3$, and $\gamma_4$ are parameters to be estimated. $\alpha_3$ and $\gamma_3$ are parameters of control variables. $\varepsilon_{4i}$ and $\varepsilon_{5i}$ are the random-error terms.

*3.4. Definition of Variables and Summary Statistics*

1. The core explained variable. The impact of land transfers on farmers' adoption of new varieties was tested with the goal of examining the benefits brought to farmers by appropriate land scale operations and UAV services. The new varieties in current research mainly fall into three categories. One of them is a plant species that has been bred artificially or cultivated based on wild plants discovered, which has novelty, specificity, consistency, and stability and has been appropriately titled. Another is a newly available variety. The third is a variety which was adopted by farmers for the first time. Xu et al. pointed out in their research on the depth and intensity of new corn varieties' adoption that whether it is a new variety is decided by farmers [64]. Even though a variety may have appeared in the market for a long time, it is a new variety for this farmer, as long as the farmer plants the variety for the first time. On the basis of comprehensive relevant studies, this study adopted the research methods of Xu Zhigang [64] and defines a new variety as "the variety adopted by the farmer for the first time, no matter how long it has been used in the market". Accordingly, the answers to the question "In the last three years, have you adopted a new variety" in the questionnaire were as follows: "yes" answers are assigned a "1", while "no" answers are assigned a "0".

2. Explanatory variable. Land transfers are conducive to improving the stability of farmland, which is the core of basic production materials. The willingness to make long-term investments such as the adoption of new varieties may increase with the perception of the land value. In the field survey questionnaire, the relevant question was expressed as "In the past three years, apart from the collective allocation of land, have you rented land from others or collectives". If the answer was "yes", then land in-transfers behavior was confirmed. That is, those who actually had land in-transfers were assigned a "1", while those who had no land in-transfers were assigned a "0".

3. Mediating variable. The model introduced the intermediary variable, which is the scale of kiwi planting, in order to test whether land transfers affect the new-variety adoption behavior of farmers through the influence of the planting scale. The research group asked the surveyed farmers "How many acres is the actual area for kiwifruit growing" The answer directly represents the level of the planting scale.

4. Moderating variable. This paper introduced the small agricultural machinery service as the moderating variable to test whether the impact of land transfers on the adoption of new varieties by farmers varies with the intensity of UAV services. On the basis of previous studies [57–59], this paper used the pollination technology of UAVs and the application of UAV spraying technology in the process of agricultural planting to measure the mechanization level of small agricultural machinery services. The relevant question in the questionnaire was as follows: "In the process of kiwifruit planting, does your family use UAVs for pollination or spraying". The answers denote the level of small agricultural

machinery services, with "1" indicating that the service of UAVs was confirmed and "0" indicating that the use of UAVs was unconfirmed.

5. Control variables. In order to reduce the error of omitted variables and avoid the interference of other factors that may affect the land transfers and the adoption of new varieties and improve the accuracy of model estimation, the characteristics of household heads, families, production, and geographical location were controlled. Individual characteristics of the household heads are age, gender, education level, and health status. The amount of family labor force, types of vehicles, and house were selected to represent the household characteristics. The distance from the house to the nearest market was measured as a geographical characteristic variable. The main variable definitions and descriptive statistics are reported in Table 1.

**Table 1.** Distribution of questionnaires.

| Variable | Variable Names | Variable Definition | | Mean | sd |
|---|---|---|---|---|---|
| Explained variables | New variety adoption | In the last three years, have you adopted a new variety | yes = 1, no = 0 | 0.207 | 0.405 |
| Explanatory variables | Land in-transfers | In the last three years, have you rented land from others or collectives | yes = 1, no = 0 | 0.224 | 0.417 |
| Mediating variable | Area of kiwifruit planting | How many acres is the actual area for kiwifruit growing | Actual area of kiwi | 5.24 | 4.58 |
| Moderating variable | UAV service | In the process of kiwifruit planting, does your family use UAVs for pollination or spraying | 0 = none of them, 1 = one or both of them | 0.392 | 0.728 |
| Characteristics of the household head | Age | Age of the head of household | Actual age | 52.12 | 8.906 |
| | Gender | Gender of the head of household | male = 1, female = 0 | 0.646 | 0.479 |
| | Education level | The highest level of education the household' head received | 1 = illiterate, 2 = primary school, 3 = junior high school (or technical secondary school), 4 = enior high school (or junior college), 5 = undergraduate and above | 2.671 | 1.188 |
| | Health | Health status of the household's head | Health level 1~5, in which level 1 is the worst health condition | | |
| Family characteristics | Labor force | The amount of labor force available | Amount of labor available | 2.433 | 1.188 |
| | House | The building type of the house | 1 = thatched cottage, 2 = earth and wood, 3 = brick and wood, 4 = brick and concrete, 5 = reinforced concrete | 0.458 | 0.498 |
| | Vehicle | Types of vehicles in the family | 1 = none, 2 = bicycle, 3 = motorcycle, 4 = car | 1.261 | 1.045 |
| Geographical location | Distance | Distance from house to the nearest market | Actual distance | 9.967 | 8.999 |

### 3.5. Summary Statistics

In general, most kiwi growers adopted more than one variety and allocated different land areas to different varieties. The level of adoption of new varieties, namely Ruiyu, Jinfu, Qihong, and Nongdayuxiang, is still relatively low, which needs to be improved. Among the 898 kiwifruit growers interviewed and surveyed, 186 households adopted the new varieties of kiwifruit, accounting for 20.7%. A total of 201 households actually had land in-transfers, while another 697 households had no land in-transfers. For details, see Table 2 below.

**Table 2.** The overall distribution of new variety adoption and land transfers.

| Type | Adopters of a New Variety | Non-Adopters of a New Variety | Total |
|---|---|---|---|
| Land in-transfers | 154 | 47 | 201 |
| No land in-transfers | 32 | 665 | 697 |
| Total | 186 | 712 | 898 |

The quantities and proportions of adopters and non-adopters of new kiwi varieties are shown in Figure 3. Among the households who adopted the new varieties, 100 of them adopted Ruiyu, accounting for 11.14% of the total. In addition, 77 households adopted Jinfu, for a proportion of 8.57% in total, and 20 households adopted Nongdayuxiang, accounting for 2.23%. Among the non-adopters, Xu Xiang and Cuixiang were mainly adopted, by 515 and 255 households, respectively. Old varieties, such as Haiwode and Qinemei, accounted for only 158 and 57 households, respectively. With the characteristics of lower technical planting requirements and easier management of the orchards compared to the new varieties, old varieties such as Haiwode, Yate, and Qinmei have higher yields but less market value, and their average commercial price has been constantly declining. In recent years, Xuxiang and Cuixiang have been widely adopted by farmers, the market prices and yields of which are much higher than those of the old varieties. However, the adoption level of improved varieties such as Jinfu, Ruiy, and Nongdayuxiang is generally low, with superior market prices but lower yields in the first few years. Even though the new varieties are favored by more consumers because of their sweet taste, disease resistance, and high quality, most farmers are reluctant to adopt new varieties due to difficult technologies and uncertain yields, which seriously restricts the improvement of farmers' income and the efficiency of the kiwi industry.

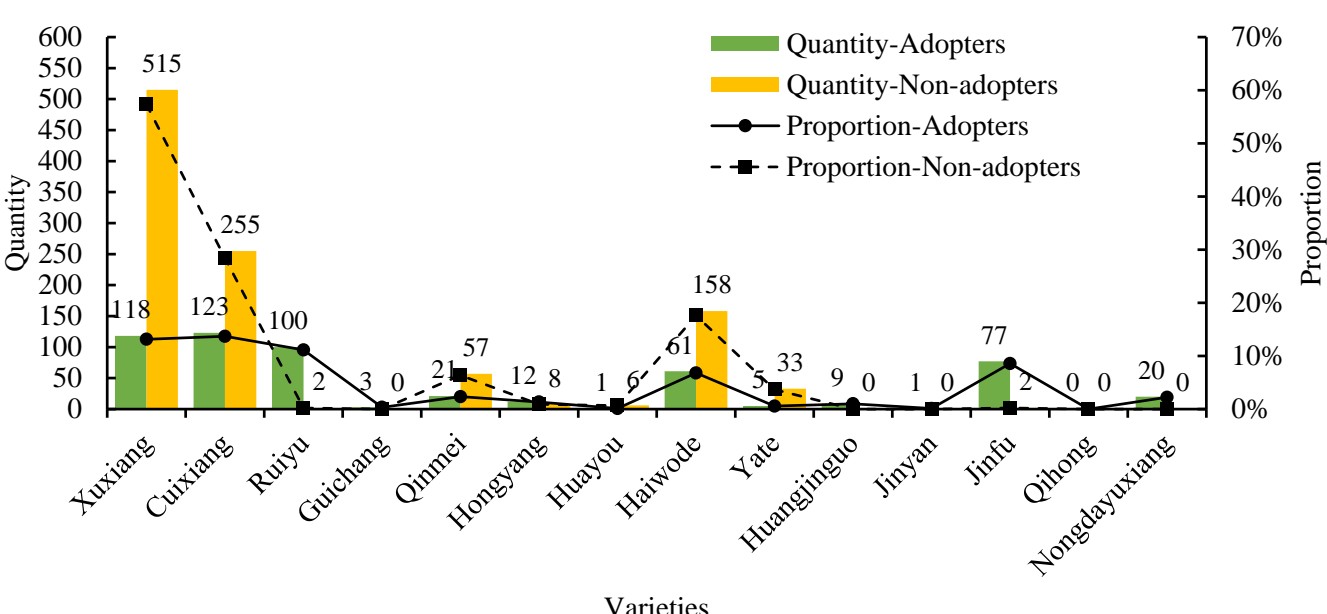

**Figure 3.** The quantities and proportions of adopters and non-adopters of new kiwi varieties.

The farmers and producers were male-dominated in terms of decision making, with 580 male individuals in total, accounting for 64.5%. The average age of the farmers and producers was 52.12 years old, with the youngest being 30 years old and the oldest being 78 years old. A total of 60 people were under the age of 40, accounting for 6.6%. The largest number of middle-aged workers were aged 40 to 60 years old, reaching 620 people, accounting for 69% of the total sample size. There were 182 people aged 60 to 70, accounting for 20.2%. There were 36 people aged 70 and above, accounting for 4% in total. It can be

seen that the current rural labor force engaged in agricultural cultivation is mainly middle-aged and older, and the aging problem is relatively serious. The education level of the sample is mainly primary school and junior high school, with 632 households, accounting for 70.3% of the total sample. The minimum value of the labor force in these households is 0, whereas the maximum number of households is 7. The main reason for this is the loss of labor competence.

The maximum values of the labor force are 2 and 3, accounting for 46.5% and 23.9%, respectively. The health level is also a factor that affects labor force input. The higher the level of health, the more labor force can be invested. However, with the development of urban–rural integration, more and more farmers are engaging in concurrent employment. Most of the young and strong laborers worked in cities, and most of the rural laborers were old women and children, which leads to a labor gap in agricultural farming. The type of housing can reflect the level of household physical capital to some extent. On the one hand, farm machinery holdings can measure the level of a household's assets. On the other hand, they represent the degree and level of participation in agricultural mechanization to a certain extent. The location of farmers may be an important factor affecting land transfers, while the number of vehicles is an important channel to communicate with the outside world and obtain various resources and information. Considering that regional economic development has a great impact on farmers' pursuit and seizure of land transaction opportunities, the distance to the nearest market was chosen to represent the geographical location characteristics of farmers. The longer the distance from the market, the less active the economy, and the more difficult and costly the land use. The distance to the nearest market was determined based on the actual location of the farmers. The distance to the nearest market in the sample varies from 0.5 km to 120 km, with an average distance of 9.97 miles.

## 4. Results

SPSS 20.0 was used to verify the multicollinearity of the independent variables. The variance inflation factor (VIF) of each independent variable was utilized to explain whether there was obvious multicollinearity between variables. Detailed results are given in Table 3.

**Table 3.** Results of the multicollinearity of the independent variables.

| Variable | VIF | 1/VIF |
|---|---|---|
| Land in-transfers | 1.69 | 0.592674 |
| Gender | 1.06 | 0.941177 |
| Age | 2.56 | 0.390855 |
| Health | 1.89 | 0.528098 |
| Education | 2.34 | 0.428106 |
| Labor force | 1.33 | 0.753724 |
| House | 1.04 | 0.965027 |
| Vehicles | 1.14 | 0.880663 |
| Distance | 1.26 | 0.790697 |
| Mean VIF | 1.59 | |

As shown in Table 3, the VIF value of age is the highest at 2.56, and that of the type of house is the lowest at 1.04, neither of which exceeds 10. Since the mean VIF of the variance inflation factor for each variable in the sample is 1.59, which is much less than 10, the principle of independence is met, and there is no severe multicollinearity. The propensity score matching (PSM) technique was then used to estimate farmers' adoption behavior toward the new varieties. The results of the tests are presented as follows.

### 4.1. Test of Co-Supporting Hypothesis and Balance Hypothesis

### 4.1.1. Co-Supporting Hypothesis

In order to ensure the matching quality and the reliability of the estimated results, it is necessary to verify the co-supporting hypothesis and the equilibrium hypothesis. Figure 4 shows the density function of the land in-transfers group and the control group after matching. It can be seen that the propensity score intervals of the post-matching group and the control group have a considerable overlap, indicating that most of the observed values are within the common value range, only a small number of samples will be lost during propensity score matching, and the common support condition will be satisfied.

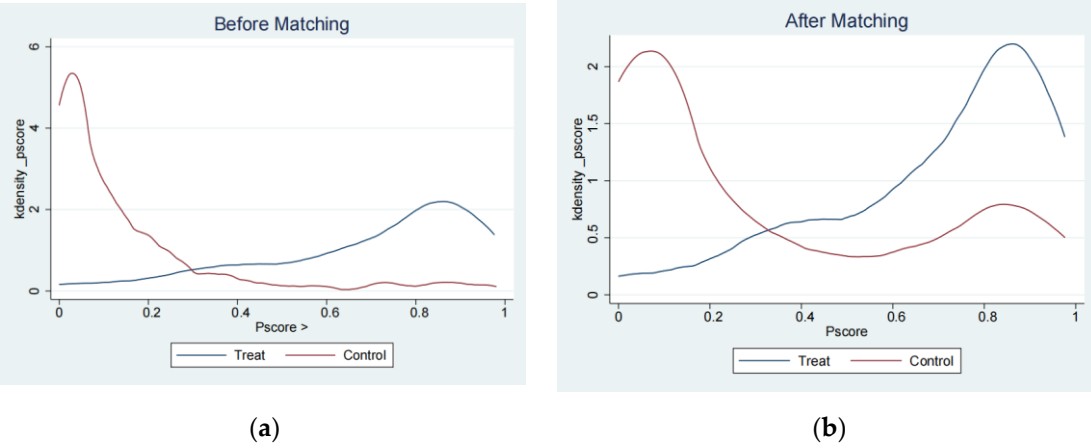

(**a**)                    (**b**)

**Figure 4.** The density function of variables' propensity scores of the land in-transfers group and the control group: (**a**) The density function graph of variables' propensity scores before matching; (**b**) The density function graph of variables' propensity scores after matching.

### 4.1.2. Test of Balance

In order to ensure the reliability of propensity score matching results, this paper tested the balance of covariables; that is, after matching, there was no significant systematic difference in covariables between the control group and the treatment group, except for the difference in the actual adoption of new varieties. This paper uses Rubin and Rosenbaum's method as a reference to test the balance in three aspects [65,66]. The standard deviation of matching variables between the treatment group and the control group before and after matching was compared. The reduction in standard deviation indicated that the difference between the two groups was reduced. The mean value of matching variables between the post-matching group and the control group was investigated to determine whether there was a difference, and the T-test was used to determine whether the difference was significant. The Pseudo-$R^2$, chi-square ($x^2$), and mean bias were examined to check whether the matching satisfied the balance assumption as a whole. Details can be found in Figures 5 and 6.

After matching, the standard deviation of each matching variable in the treatment group and the control group was significantly reduced, and the total bias was significantly reduced, being less than the 20% red line standard stipulated by the balance test, indicating that the sample matching was relatively successful.

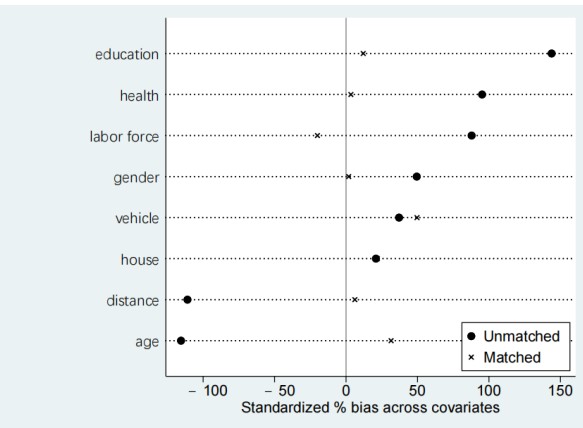

**Figure 5.** Equality test of the means of variables before and after matching.

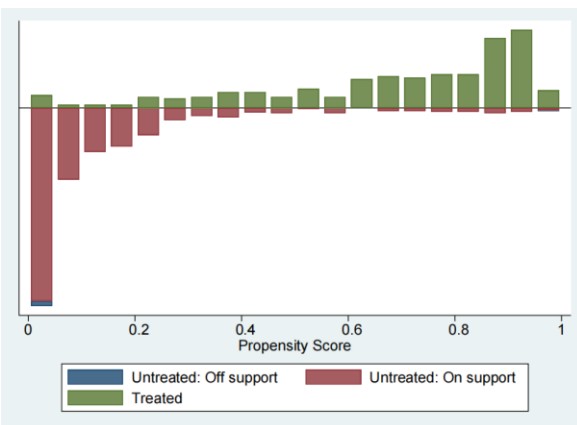

**Figure 6.** Propensity test score distribution and common support. Treated: On support indicates adopters of new kiwi varieties that had a suitable comparison group (non-adopters). Treated: Off-support indicates adopters of new kiwi varieties that did not have a suitable comparison group (non-adopters). Source: field survey, 2020.

*4.2. Estimation and Analysis of Propensity Score Matching*

Based on the propensity score value, this paper matches the samples and calculates ATT and ATU under the two situations of farmers' land transfers and non-transfers, respectively. Commonly used matching methods include nearest K-nearest neighbor matching, kernel matching, caliper matching, and spline matching. There was no significant difference in the results obtained using these four matching methods in this paper, which is consistent with the conclusion that matching methods have little influence on matching results in existing studies. Therefore, this paper mainly discusses the estimation results of kernel matching.

Table 4 describes the results of the PSM test. After counterfactual estimation with propensity score matching, the ATT-estimated value of the treatment group (i.e., kiwifruit growers with land in-transfer behavior) was 0.125 and was significant at the level of 5%, indicating that land transfers had a positive and significant impact on the actual behavior of kiwifruit growers in adopting new varieties. When the net effect was not affected by land transfer behavior (counterfactual), the adoption behavior of new varieties significantly increased by 12.5%, which is generally consistent with the above findings; thus, hypothesis 1 was verified.

**Table 4.** Results of the PSM test.

|  | Observed Coef. | Bootstrap Std. Err. | **P > |z|** |
|---|---|---|---|
| ATT | 0.125 ** | 0.058 | 0.029 |

Source: field survey, 2020. ** denotes significance at 5%.

### 4.3. Results of Mediating Effects

In accordance with the above analysis, the stepwise regression method was used to explore the potential mechanism of how land in-transfers affect the adoption of new varieties. Model 1 explains the direct effect of land in-transfers on the adoption of new varieties. The impact of land in-transfers on the kiwi area is shown in model 2. Model 3 explains the influence of land in-transfers on the adoption of new varieties after the mediating variable (kiwi area) is introduced. Detailed results are given in Table 5.

**Table 5.** Results of the benchmark model and mediating effects.

| Variable | Model 1 | Model 2 | Model 3 |
|---|---|---|---|
| Land in-transfers | 0.472 *** (19.316) | 3.507 *** (8.973) | 0.417 *** (16.875) |
| Gender | −0.000 (−0.003) | 0.960 *** (3.550) | −0.015 (−0.913) |
| Age | 0.004 ** (2.758) | −0.072 ** (−3.210) | 0.005 *** (3.653) |
| Health | −0.009 (−0.879) | −0.210 (−1.315) | −0.005 (−0.567) |
| Education | 0.140 *** (12.481) | 0.327 (1.822) | 0.135 *** (12.389) |
| Labor force | 0.054 *** (7.109) | 0.406 *** (3.341) | 0.048 *** (6.435) |
| House | 0.012 * (2.217) | 0.426 *** (4.739) | 0.006 (1.053) |
| Vehicles | 0.001 (0.081) | 0.245 * (2.052) | −0.003 (−0.445) |
| Distance | −0.003 ** (−3.169) | −0.009 (−0.564) | −0.003 ** (−3.125) |
| Area of kiwi | — | — | 0.016 *** (7.687) |
| _cons | −0.588 *** (−5.245) | 5.137 ** (2.866) | −0.668 *** (−6.125) |
| Sample size | 898 | 898 | 898 |

Source: field survey, 2020. ***, **, and * denote significance at 1%, 5%, and 10%.

As can be seen from model 1, land transfers significantly increased the adoption of new varieties by farmers after controlling for basic characteristic factors of the household head, being significant at the 5% confidence level. This result is consistent with the PSM test. Age, education level, labor force, and housing type all had a significant positive response to the adoption of new varieties. On one hand, the older the farmers are, the more years they have been involved in farming and thus have accumulated more experience in farming, which makes it easier to master the techniques related to using new varieties. On the other hand, the older one is, the less willing one may be to take the risk of adopting new varieties with uncertain yields and uncertain technical mastery. Farmers with low levels of education are less willing to participate in the adoption of new varieties, while farmers with higher levels of education are more willing to participate. Possible reasons for this are that the higher the level of education, the more knowledgeable a person is, and the more confident and capable they are in mastering and using the technology related to new varieties. The richer the labor force, the higher the productivity and the more likely it is to be transferred to the land for large-scale cultivation. Distance to the nearest market has a

significant negative impact on farmers' adoption of new varieties. The farther away from the market, the less developed the economy is likely to be, and the more limited access to and availability of information about new varieties are, which may have a negative impact on farmers' adoption of new varieties. However, gender, health level and vehicle ownership did not have significant effects on the adoption of new varieties in this model.

Model 2 indicates that land transfers had a significant positive contribution to the planted area of kiwifruit at the confidence level of 1%. When the value of land transfers is 1, the planting area of kiwifruit increases by 3.507 units. Gender, education level, number of laborers, housing type, and vehicle had a positive effect on the kiwifruit planting area, while age had a negative effect on the kiwifruit planting area.

Model 3 reports the results of the mediation test. The planting area of kiwifruit plays a positive mediating role in the process of land transfers, affecting farmers' adoption of new varieties, with an impact coefficient of 0.016, which was significant at the 1% level. After introducing the kiwifruit area as a mediating variable, the effect of land transfers on the adoption of new varieties remained significantly positive. However, the coefficient values decreased compared to the corresponding coefficients in model 1, indicating that the kiwi area played a partially mediating role in the relationship between land transfers and new variety adoption. Hypothesis 2 was verified.

### 4.4. Results of the Moderating Effect

In the test of the moderating effect, the core independent variable and the moderating variable were standardized. As can be seen from Table 6, the moderating effect was divided into two models. Besides the core independent variable (land in-transfers) and core dependent variable (new variety adoption), model 4 and model 5 adopted the same control variables mentioned above, including age, gender, education level, health, labor force, housing type, vehicle, and distance from the house to the nearest market. It can be seen that model 4 is based on model 1, with the introduction of a moderating variable (UAV service), and model 5 is based on model 4, with the introduction of an interaction term (the interaction between land in-transfers and UAV service). Table 6 shows the detailed results of the moderating effect.

For model 4, the objective was to investigate the effect of the independent variable land in-transfers on the adoption of new products as the dependent variable, without considering the interference of the moderating variable. The estimates of the land transfer parameter were positive and significant at the 1% level. The results indicate that land transfers have a significant positive effect on the adoption of new varieties by farmers. The magnitude of the coefficients of the variables varied slightly due to the different methods used in the model, but the direction and significance of the variables were consistent with the above, and the model was relatively robust.

The moderating effect of UAV service was remarkable (Table 6 and Figure 7). Figure 7 illustrates the relationship between land in-transfers and the adoption of new varieties. The slope visually shows the difference in the effect of land transfers on farmers' adoption of new varieties when the service of UAVs was at different levels. The three levels of the control variables are as follows: the average level; the high level, which represents the average plus one standard deviation; and the low level, which represents the average minus one standard deviation. When the moderating variable is whether to use UAVs to take different levels, the difference in the magnitude (slope) of the effect of the independent variable land transfer on the dependent variable of whether farmers adopt new varieties is the specific moderating effect. The slopes of the three differ and finally intersect, indicating that the interaction term does have a moderating effect on the effect of land transfers on farmers' adoption of new varieties.

**Table 6.** Results of the moderating effect.

| Variable | Model 4 | Model 5 |
|---|---|---|
| Land in-transfers | 0.183 *** | 0.126 *** |
| | (18.305) | (11.503) |
| Gender | −0.015 | −0.009 |
| | (−0.903) | (−0.591) |
| Age | 0.003 * | 0.001 |
| | (2.531) | (0.688) |
| Health | −0.013 | −0.002 |
| | (−1.382) | (−0.238) |
| Education | 0.115 *** | 0.092 *** |
| | (10.175) | (8.467) |
| Labor force | 0.047 *** | 0.052 *** |
| | (6.359) | (7.419) |
| House | 0.009 | 0.006 |
| | (1.633) | (1.132) |
| Vehicles | 0.002 | −0.013 |
| | (0.276) | (−1.889) |
| Distance | −0.001 | −0.002 * |
| | (−1.278) | (−2.105) |
| Unmanned aerial vehicles (UAVs) | 0.079 *** | 0.070 *** |
| | (7.886) | (7.322) |
| Land in-transfers × Unmanned aerial vehicles (UAVs) | — | 0.089 *** (10.557) |
| _cons | −0.360 ** | −0.214 * |
| | (−3.276) | (−2.042) |
| Sample size | 898 | 898 |

Source: field survey, 2020. ***, **, and * denote significance at 1%, 5%, and 10%. Land in-transfer×Unmanned refers to the interaction term of Land in-transfers and Unmanned aerial vehicles (UAVs).

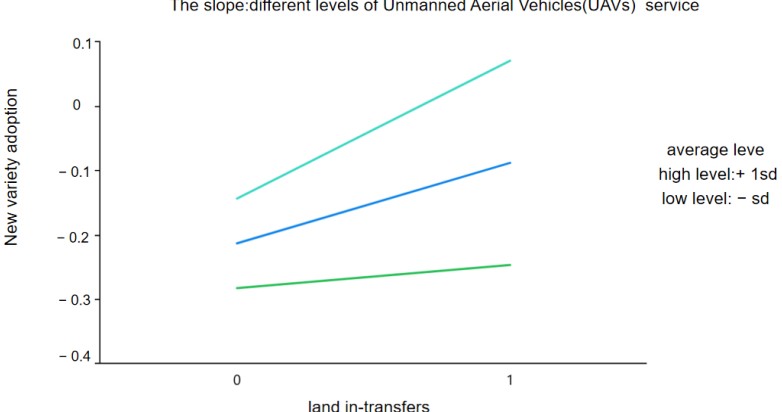

**Figure 7.** The relationship between land in-transfers and the adoption of new varieties.

Looking specifically at the regression results of model 5, the interaction coefficient of land in-transfers and UAVs is 0.089, which is positive and significant at the 1% level. This indicates that the service of UAVs play a positive moderating role when land transfers affect the adoption of new varieties by farmers. The effect of land transfers on farmers' adoption behavior differs from the intensity of UAV services. Hypothesis 3 was verified.

*4.5. Robustness Test*

The endogenous transformation probit (ESP) model was used for the robustness test. As the existence of local land transfer support policies is a strictly exogenous variable, it may directly affect land transfer, but does not necessarily affect the adoption behavior of farmers. Therefore, the government support policies for land transfers are selected as the

instrumental variable. According to the estimation results of the instrumental variables model (IV-2SLS), the F value of the first stage was 29.17, indicating that the instrumental variables were not weak. The results of the endogenous conversion probit model showed that the correlation coefficients $rho^1$ and $rho^0$ of the error terms estimated by the land transfer model and the new variety adoption model were statistically significant at the 1% and 10% confidence level, respectively, indicating that the new variety adoption model did have sample selection bias, which was consistent with the theoretical analysis above. The Wald test values of equation independence are 17.02 and 14.53, respectively, both of which reject the null hypothesis that the selection equation and the result equation are independent of each other at the confidence level of 1%. The goodness of fit test of the model was significant at the level of 1%, and the ATT value of the main effect was 0.19 and significant at the 1% confidence level, indicating that land transfers had a significant positive impact on the adoption of new varieties by farmers, which was consistent with the conclusions above; thus, the robustness was verified.

## 5. Discussion

### 5.1. Evaluation of the Impact of Land Transfers on the Adoption of New Varieties

Based on the survey data of 898 kiwifruit growers in Shaanxi Province, this paper analyzed the effect of land transfers on the adoption of new varieties. To address concerns regarding the presence of some unobservable factors that may affect the results and self-selection performance of farmers, propensity score matching (PSM) technology was utilized to solve these problems. The results indicate that land in-transfers can significantly improve the level of adoption of new varieties. To be more specific, farmers with land transfers had a significant positive effect on the adoption of new kiwifruit varieties, and the net effect of land transfer significantly increased by 12.5% when there was no land transfer behavior (counterfactual). The endogenous transformation probit model was used to test the robustness of the empirical models, which showed basically identical results of the PSM test. The basic benchmark regression results also show that land transfers positively promoted the adoption of new varieties by farmers. The reasons accounting for the results can be explained as follows. For one thing, farmers with land transfers tend to have more accumulated assets, a higher education level, richer farming experience, and stronger information-acquisition and bargaining power, which may result in a better understanding and recognition of new varieties of technologies. Such farmers are able to broaden the distribution channels of new kiwifruit through cooperation with agricultural enterprises and e-commerce, which helps to increase the awareness of the market value of the new varieties. For another, land in-transfers proved to be an effective way to concentrate fragmented and scattered land for professional growers, who are more inclined to take the risk of adopting new varieties and making long-term investments in land. Last but not least, land in-transfers can lead to farmland expansion, which makes it easier for farmers to meet the conditions required for implementing mechanical operations and adopting new agricultural techniques to reduce labor costs and improve production efficiency.

### 5.2. Mediating Effects of Land Scale Operations

The kiwi planting area plays a partially positive mediating role, suggesting that land in-transfers enhance the adoption of new varieties by way of increasing the land scale, which generates land scale effects. The positive coefficients of education and labor force also demonstrate that households with a higher education level and more capable labor force are more inclined to adopt new varieties. According to the analysis of general knowledge, land transfers directly increase farmers' land holdings, which is conducive to increasing the level of farmers' perception of land value, directly influencing the level of agricultural operation and facilitating the formation of the scale effect. The adoption of new varieties is a risky behavior of long-term investment which requires risk tolerance and risk resistance. Moderate scale operations are more conducive to farmers' long-term, stable positive perception of new varieties and land value, stimulating their intention of

making long-term investments in land and making them willing to adopt new and superior varieties with a higher market value and thus obtain higher returns.

### 5.3. Moderating Effects of UAV Services

The services of UAVs played a positive moderating role in the relationship between land transfers and the adoption of new varieties. When the services of UAVs were at different levels, there were significant differences in land transfers, affecting the adoption of new varieties. The use of small agricultural machines such as UAVs can improve pollination accuracy and spraying levels and also reduce the labor costs brought about by a scale up. Thus, small farm machinery services and land transfer behavior are complementary and mutually reinforcing. This is because farmers with land transfer behavior may have stronger household wealth accumulation and more combined capabilities than those without land transfers. For instance, they have a higher household income, richer social capital, stronger confidence in land management, and greater self-efficacy, and are more likely to engage in large-scale operations and small-scale farm machinery operations. Large cultivators with land transfers usually have stronger negotiation and resource-acquisition abilities. They can cooperate with rural cooperatives, supply and marketing societies, and agricultural technology service centers to obtain more information and technical services from new technologies and reduce the input costs and risk perceptions related to the adoption of new varieties.

### 5.4. Limitations and Future Directions

From the theoretical perspective, this research contributes to the current literature by revealing the relationship between land in-transfers and the adoption of new varieties. Although land transfer policies have been in place in China for many years, their impact on the adoption of new varieties has not been clearly explained. Empirical studies have been carried out to determine the effect of land transfers on scale and agricultural output [46,48]. However, there is little in the literature on the impact of land in-transfers on the adoption of new varieties. Compared with Cao et al. [45], this study explored the underlying mechanism by verifying the mediating role of kiwi area and identifying the moderating effect of UAVs between land in-transfers and the adoption of new varieties. This study has highlighted the positive influence of land in-transfers on the adoption of new varieties. On one hand, the mediating role of kiwi area implies the importance of contiguous land. On the other hand, the examination of moderating factors further reveals that intensity varies with different levels of UAV service. These findings provide us with a deeper understanding of how land in-transfers affect the adoption of new varieties, which provides an empirical basis for policymakers and other participants in agricultural production. Cultivated land in China has a high degree of fragmentation, which leads to a large amount of small-scale farm operations. The results of this paper can also be applied to other cash crops and some developing countries in Asia where small-scale households represent the majority of agricultural producers.

This study also has some limitations which require future research. First, the study used cross-sectional data for just one year of 2020. Nevertheless, the process of land transfers and the adoption of new varieties may require a much longer period. The same households are hard to follow to ensure consistency with the previous field survey. Second, besides other kiwi-producing areas such as Sichuan, Guizhou, and Hubei provinces, the main production base of Shaanxi Province is selected as the representative sample, which may result in the conclusion not being fully representative. Third, household heterogeneity is rarely discussed in this paper. Due to the heterogeneity of resource endowments, the effect of land transfers on the adoption of new varieties may vary between different types of households, such as ordinary small-scale farmers, family farms, and professional growers. Future studies can be carried out when more information from field surveys over longer periods is acquired, and the sample is expanded over different regions. The mechanism of land transfers affecting the adoption of new varieties due to household heterogeneity can also be explored, so as to strive for more contributions to enrich the research field.

## 6. Conclusions and Recommendations

The land is the foundation of human activities and both economic and social development and is a precious resource that impoverished, small farm households rely on for survival. Land transfers are an effective way to reduce land fragmentation and improve land efficiency. The adoption of new varieties seeks to promote land productivity and the quality of agricultural products. It also aims to increase the growth of farmers' incomes and narrow the gap between the rich and poor. This research has highlighted the positive influence of land in-transfers on the adoption of new varieties based on an empirical study of new-variety adoption by kiwi farmers in Shaanxi, China. A field survey of the kiwi households was carried out, and the PSM method was applied to analyze the different attitudes towards new varieties between the adopters and non-adopters. In addition to this, the mediating role of kiwi area and moderating role of UAVs were also emphasized. The results suggest that kiwi area plays a significant mediating role in the process, while UAV services have a positive moderating effect on the impact of land in-transfers on the adoption of new varieties, illustrating the importance of land scale operations and small agricultural machinery services. From that presented above, the following suggestions and implications are proposed to raise the level of land use efficiency and achieve an improvement in the adoption of new varieties on the basis of this research. For the kiwifruit industry, the adoption of new varieties and the application and promotion of new technologies is the general trend towards achieving high-quality development and improving management efficiency. However, the current situation of farmers using new and better varieties is still not optimistic, as most small farmers have a conservative mindset. Such farmers prefer using the old varieties to secure the harvest in droughts and floods and are not willing to try new and better varieties. The new breeds adopted by new business entities are also relatively limited in scale. The renewal of kiwifruit varieties is essentially a market behavior, but the market value of some old varieties is declining, which is not conducive to sustainable income growth for farmers. Faced with the current situation, in which the level of land transfers and the adoption of new varieties is generally low, the government should take active measures to guide farmers to adopt new and superior varieties to improve the economic added value of agricultural products, increase farmers' income, and narrow the gap between the rich and poor in rural areas.

First, since land in-transfers have been proved to be significantly positive in affecting the adoption of new varieties, the institutional construction of land ownership confirmation and certification needs to be further improved, as well as the market trading mechanism of land property rights, which aims to make land management rights easier to trade between small farm households and other agricultural enterprises. Clear land use rights are the premise of land efficiency, ensuring the interests of small farm households. Therefore, the government should solidify the ownership of contractual management rights and create sound conditions to reduce the obstacles and costs of the land transaction market.

Second, as the land scale has a partial intermediary effect on the influence of land in-transfers on the adoption of new varieties, the government and agricultural extension centers should focus on reducing the degree of farmland fragmentation and land integrity in order to generate scale effects, which increase the adoption of new varieties. It is necessary to establish incentive policies to guide small farmers to make full use of land and reduce the area of abandoned land in order to accelerate the shift of agricultural production towards specialization and a moderate scale. Related legal protection for land transfers should be provided to integrate the land resources and realize the highest level of land use utility.

Third, according to the estimation results of the moderating effects of UAVs, joint efforts from the government, agricultural enterprises, and rural professional cooperatives are required to promote agricultural machinery services, which can help small households to effectively participate in modern agricultural production. The professional cooperatives should make full use of their advantages in purchasing agricultural machinery services and agricultural materials to reduce the production costs of new varieties. In terms of small agricultural machinery services, the advantages of UAVs should be fully publicized among

farmers so that they can more actively understand and adopt this new technology, which can not only improve the quality of pollination and reduce the use of pesticides, but also reduce the cost of manual operations. In addition, the use of UAVs can also achieve precise spraying and positioning, thus making tremendous contributions to green agricultural development and sustainable land use.

Finally, due to the positive coefficients of education and labor force, an increase in the special and frequent training of new-variety planting techniques is required to improve the capabilities of small farm households. The publicity of new and superior varieties also needs to be enhanced, especially with regard to the successful operation of new varieties, which can greatly improve the confidence of small farm households. Timely price information and market information should be provided to make the new varieties more attractive to farmers. Periodic expert consultations for farmers need to be formulated to reduce uncertain planting risks in the production of new varieties. Guidance on diversified forms of internet use, such as e-commerce platforms, is encouraged to expand sales channels, publicity, and influence to empower the adoption of new varieties, thereby boosting the overall general increase in farmers' income.

**Author Contributions:** Conceptualization, Y.C.; methodology, Y.C.; software, Y.C.; validation, Y.C.; formal analysis, Y.C.; investigation, Y.C.; resources, Y.C.; data curation, Y.C.; writing—original draft preparation, Y.C.; writing—review and editing, Y.C.; visualization, Y.C.; supervision, Z.W.; project administration, Z.W.; funding acquisition, Z.W. All authors have read and agreed to the published version of the manuscript.

**Funding:** This research was funded by the Shaanxi Philosophy and Social Science Research Key Project entitled "Ecological Space Governance in Shaanxi Province", 2022, China (Grant No: 2022HZ1764); The Yulin Science and Technology Project entitled "Key technology selection and economic benefit evaluation of apple industry value chain under the background of common prosperity", 2022, China (Grant No: CXY–2022–60).

**Data Availability Statement:** The data used in this study contain sensitive information about the study participants, and they did not provide consent for public data sharing. A minimal data set could be shared upon request by a qualified academic investigator for the sole purpose of replicating the present study.

**Conflicts of Interest:** The authors declare that they have no conflict of interest to report regarding the present study.

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
