# Peer review of "The Impact of Land Transfers on the Adoption of New Varieties: Evidence from Micro-Survey Data in Shaanxi Province, China"

_land, doi:10.3390/land12030632_

Round 1

Reviewer 1 Report

The subject is relevant and interesting. The structure of the model is clear, easy to construct and well described. The argumentation and methodology in the paper is convincing. I much like the scientific and logical layout and flow of this paper leading to the conclusions. The nature of this paper was such that it needs to be understandable and accessible both to academic audiences and policymakers. 

Author Response

Dear Reviewer 1,

We would like to express our appreciation to you for the overall very positive assessment of our manuscript.All the comments have been studied very carefully and the corresponding changes have been made.Please see the attachment for detailed response.

Kind regards,

Yi Chen

Reviewer 2 Report

Dear Authors,
This is an interesting manuscript with subtle and very useful solutions, however there are some aspects that should be improved, such as the following:
52. When you mention "many scholars have conducted...", you should cite several of them.
I noticed that the references in the text have no reference number. For example, line 55, 60. the entire manuscript should be corrected.
60. insert name of article, paper, or analysis by Nazli.
84. "Recently, some studies..." please detail three of them.
126. "Existing studies have shown that farmers..." please list several of these studies.
143. "The previous studies show..." please detail. The same applies to lines 179, 188, and 194. Check the references if you list more than one study... References must be coherent, if that.
Figure 1 is not linked in the text. Also, it does not explain anything. It needs to be improved or removed.
211 - 215. please reference the data.
212. values with more than 5 digits should be separated with ",".
Figure 2. is not referenced in the text. You can add the kiwifruit planting information to the map. Or you can create another map.
264 - 270. Insert the data source from institutions, universities, or agencies that generate detailed information.
277 - 280. keep the same format between text and formulas.
314. improve writing, "We study..." and change the first person form. Review the entire manuscript.
There is a lack of general methodology. In Tables 4 and 5, you suddenly mention models 1, 2, 3, 4, and 5 without prior and supported introduction. It is necessary to include a logical methodology linked to the hypothesis, data sources, models (3.3. Definition of variables and summary statistics), validations and results.
The manuscript needs to be restructured in terms of methodology, results, discussion, and conclusions. For example, there is good discussion and conclusion material in the results section. In doing so, the discussion and conclusions sections need to be significantly improved and supported.
Very good support because you used a logistic model, pros and cons, based on your data.
More support because you select New variety adoption as an explanatory variable, Land Transfer-in as Explanatory variables, Area of kiwi fruit planting as Mediating variable, UAVs service as Moderating variable...
Extend the results of multicollinearity of the independent variables, conduct further analysis.

If you have primary data on kiwifuit farmers, you need to analyze it in more detail. For readers: If you mention "...New Varieties' Adoption" based on 1000 kiwifruit farmer households of smallholders (kiwifruit farmers only), the results could be biased to Kiwifuit planting reality. Please substantiate the data source conditions for the results.
You must include an analysis where kiwifruit information is the Explained variable.

Author Response

Dear Reviewer 2,

We would like to express our appreciation to you  for these helpful suggestions on how we may improve this manuscript. All the comments have been studied very carefully and the corresponding changes have been made.Please see the attachment for detailed response.

Kind regards,

Yi Chen
